# K-level Reasoning for Zero-Shot Coordination in Hanabi

**Brandon Cui**
Facebook AI Research
bcui@fb.com

**Hengyuan Hu**
Facebook AI Research
hengyuan@fb.com

**Luis Pineda**
Facebook AI Research
lep@fb.com

**Jakob N. Foerster** *
University of Oxford
Jakob.foerster@eng.ox.ac.uk

## Abstract

The standard problem setting in cooperative multi-agent settings is *self-play* (SP), where the goal is to train a *team* of agents that works well together. However, optimal SP policies commonly contain arbitrary conventions ("handshakes") and are not compatible with other, independently trained agents or humans. This latter desiderata was recently formalized by [18] as the *zero-shot coordination* (ZSC) setting and partially addressed with their *Other-Play* (OP) algorithm, which showed improved ZSC and human-AI performance in the card game Hanabi. OP assumes access to the symmetries of the environment and prevents agents from breaking these in a mutually *incompatible* way during training. However, as the authors point out, discovering symmetries for a given environment is a computationally hard problem. Instead, we show that through a simple adaption of k-level reasoning (KLR) [7], synchronously training all levels, we can obtain competitive ZSC and ad-hoc teamplay performance in Hanabi, including when paired with a human-like proxy bot. We also introduce a new method, synchronous-k-level reasoning with a best response (SyKLRBR), which further improves performance on our synchronous KLR by co-training a best response.

## 1 Introduction

Research into multi-agent reinforcement learning (MARL) has recently seen a flurry of activity, ranging from large-scale multiplayer zero-sum settings such as StarCraft [35] to partially observable, fully cooperative settings, such as Hanabi [1]. The latter (cooperative) setting is of particular interest, as it covers human-AI coordination, one of the longstanding goals of AI research [10, 6]. However, most work in the cooperative setting—typically modeled as a Dec-POMDPs—has approached the problem in the *self-play* (SP) setting, where the only goal is to find a team of agents that works well together. Unfortunately, optimal SP policies in Dec-POMDPs commonly communicate information through *arbitrary* handshakes (or conventions), which fail to generalize to other, independently trained, AI agents or humans at test time.

To address this, the *zero-shot coordination* setting [18] was recently introduced, where the goal is to find training strategies that allow *independently trained agents* to coordinate at test time. The main idea of this line of work is to develop learning algorithms that can use the structure of the Dec-POMDP itself to independently find mutually compatible policies, a necessary step towards human-AI coordination. Related coordination problems have also been studied by different communities, in

---

*Work done while at Facebook AI Research

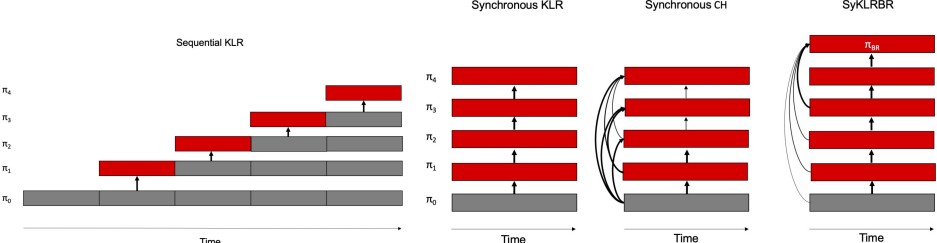

Figure 1: Visualization of various hierarchical training schemas, including sequential KLR, synchronous KLR, synchronous CH, and our new SyKLRBR for 4 levels. Thicker arrows indicate a greater proportion of games played with the level. Additionally, red boxes indicate an actively trained agent, while grey boxes indicate a fixed agent. Typically $\pi_0$ is a uniform random agent.

particular behavioural game theory. One of the best-known approaches in this area is the cognitive-hierarchies (CH) framework [4], in which a hierarchy of agents is trained. For this method, an agent at level-$k$ models other agents as coming from a distribution up to level $k - 1$ and best-responds accordingly. The CH framework has been shown to model human behavior in games for which equilibrium theory does not match empirical data [4]; thus, in principle, the CH framework could be leveraged to facilitate human-AI coordination in complex settings. A specific instance of CH that is relevant to our work is K-level reasoning (KLR) [7], wherein the level-$k$ agent models the other agents as level-$(k - 1)$. However, KLR, like many of the ideas developed in these works, has not been successfully scaled to large scale coordination problems [18].

In this paper we show that k-level reasoning can indeed be scaled to large partially observable coordination problems, like Hanabi. We identify two key innovations that both increase training speed and improve the performance of the method. First, rather than training the different levels of the hierarchy sequentially, as would be suggested by a literal interpretation of the method (as was done as a baseline in [18]), we instead develop a synchronous version, where all levels are trained in parallel (see figure 1). The obvious advantage is that the wall-clock time can be reduced from linear in the number of levels to constant, taking advantage of parallel training. The more surprising finding is that synchronous training also acts as a regularizer on the policies, stabilizing training.

The second innovation is that in *parallel* we also train a best response (BR) to the entire KLR hierarchy, with more weight being placed on the highest two levels. This constitutes a hybrid approach between CH and KLR, and the resulting BR is our final test time policy. Our method, synchronous-k-level reasoning with a best response (SyKLRBR), obtains high scores when evaluating independently trained agents in *cross-play* (XP). Importantly, this method also improves ad-hoc teamplay performance, indicating a robust policy that plays well with various conventions.

Lastly, we evaluate our SyKLRBR agents paired with a *proxy human policy* and establish new state-of-the-art performance, beating recent strong algorithms that, in contrast to our approach, require additional information beyond the game-provided observations [18, 17].

Our results show that indeed KLR can be adapted to address large scale coordination problems, in particular those in which the main challenge is to prevent information exchange through arbitrary conventions. Our analysis shows that synchronous training regularizes the training process and prevents level-$k$ from overfitting to the now *changing* policy at level-$k - 1$. In contrast, in sequential training each agent *overfits* to the *static* agent at the level below, leading to arbitrary handshakes and brittle conventions. Furthermore, training a best response to the entire hierarchy improves the final ZSC performance and robustness in ad-hoc settings. This is intuitive since the BR can carry out *on-the-fly adaptation* in the ZSC setting.

Our results show that the exact graph-structure used, which were similarly studied in [13], and the type of training regime (synchronous vs sequential) can have a major impact on the final outcome when adapting ideas from the cognitive hierarchy literature to the deep MARL setting. We hope our findings will encourage other practitioners to seek inspiration in the game theory literature and to scale those ideas to high dimensional problems, even when there is precedent of unsuccessful attempts in prior work.

## 2 Related Work

A significant portion of research in MARL has been focused on creating agents that do well in fully cooperative, partially observable settings. A standard approach is through variations of self-play (SP) methods [9, 8, 12, 24, 15]; however, as shown in [18] generally optimal SP agents learn highly *arbitrary policies*, which are incompatible with independently trained agents. Clearly, test-time coordination with other, independently trained agents including humans is an important requirement for AI agents that is not captured in the SP problem setting. To address this, [18] introduced the *zero-shot coordination* (ZSC) setting, where the explicit goal is to develop learning algorithms that allow independently trained agents to collaborate at test time.

Another recent area of work trains an RL agent separately, and then evaluate its performance in a new group of AI agents or humans assuming access to a small amount of test-time data [24, 33]. These methods build SP policies that are compatible with the test time agents by guiding the learning to the nearest equilibria [24, 33]. Other methods use human data to build a human model and then train an approximate best response to this human model, making it compatible with human play [5]. While this presents a promising near-term approach for learning human-like policies in specific settings where we have enough data, it does not enable us to understand the fundamental *principles* of coordination that lead to this behavior in the first place, which is the goal of the ZSC setting. Our paper shows that even relatively simple training methods can lead to drastically improved ZSC performance when combined with modern engineering best practices and, importantly, that the performance gains directly translate into better coordination with a human-like proxy and ad-hoc teamplay (without requiring human data in the training process).

Our approach for scaling KLR to Hanabi relies heavily on the parallel training of all of the different levels, where each level is trained on one machine and the models are exchanged periodically through a central server. This architecture draws inspiration from population based training (PBT), which was first popularized for hyperparameter turning [20] and then applied in the multi-agent context to train more robust policies in two player zero-sum team settings [19]. PBT has also been used to obtain better self-play policies in Hanabi, both in [12] and [1]. In contrast to prior work, we do not use PBT to avoid local optima or train less exploitable agents but instead leverage this framework to implement a KLR and a best response to this KLR that is geared towards ZSC and coordination with human-like proxies.

There are a few other methods directly addressing the ZSC framework. The first, *other-play* (OP) [18] requires access to the *ground truth symmetries* of the problem setting and then learns policies that avoid breaking these symmetries during training. OP has previously been applied to Hanabi and KLR compares favorably to the OP results (see Section 4). We also note, that KLR does not require access to the symmetries and can be applied in settings where no symmetries are present. The next method, *Ridge Rider* (RR) [27] uses the connection between symmetries in an environment and *repeated eigenvalues of the Hessian*, to solve ZSC problems. Like KLR, RR does not require prior ground truth access. However, unlike KLR, RR is extremely computationally expensive and has not been scaled to large scale RL problems. Life-Long Learning (LLL) has been studied for ZSC [26]. However, LLL requires access to a pool of pre-trained agents, and in this case they had access to symmetries, whereas our method never required access to such symmetries and our method compares favorably in the ZSC setting. Lastly, *Off-Belief Learning* (OBL) [17] has been shown to provide well-grounded play in hanabi and strong results in the ZSC setting, but requires *simulator access* to train. We note that KLR doesn't require simulator access and also *matches* or even *outperforms* OBL on various metrics.

## 3 Background

### 3.1 Dec-POMDPs

This work considers a class of problems, Dec-POMDPs [2], where $N$ agents interact in a partially observable environment. The partial observability implies that every agent $i \in \{1, \cdots, N\}$ has an observation $o_t^i = O(s_t)$ obtained from via the observation function $O$ from the underlying state $s_t \in \mathcal{S}$. In our setting, at each timestep $t$ the acting agent $i$ samples an action $u_t^i$ from policy $\pi_i$, $u_t^i \sim \pi_\theta^i(u^i | \tau_t^i)$, where $\theta$ are the weights of the neural networks, and all other agents take no-op actions. Here we use *action-observation histories (AOH)* which we denote as $\tau^i = \{o_0^i, u_0^i, r_1 \cdots, r_{T-1}, o_T^i\}$,

where $T$ is the length of the trajectory, and $r_t$ is the common reward at timestep $t$ defined by the reward function $R(s, u)$. The goal of the agents is to maximize the total reward $\mathbb{E}_{\tau \sim P(\tau|s,u)}[R_t(\tau)]$; here we consider $R_t(\tau)$ to be the discounted sum of rewards, i.e. $R_t(\tau) = \sum_{t'=t}^{\infty} \gamma^{t'-t} r_{t'}$, where $\gamma$ is the discount factor. Additionally, the environments in this work are turn-based and bounded in length at $t_{max}$.

## 3.2 Deep Multi-Agent Reinforcement Learning

Deep reinforcement learning has been applied to a multitude of multi-agent learning problems with great success. Cooperative MARL is readily addressed with extensions of Deep Q-learning [25], where the Q-function is parameterized by neural networks to learn to predict the expected return based on the current state $s$ and action $u$, $Q(\tau_t, u_t) = \mathbb{E}_{\tau \sim P(\tau|s,u)} R_t(\tau)$. Our work also builds off of state of the state of the art algorithm Recurrent Replay Distributed Deep Q-Network (R2D2) [21]. R2D2 also incorporates other recent advancements such as using a dueling network architecture [36], prioritized replay experience [29], and double DQN learning [34]. Additionally, we use a similar architecture as the one proposed in [14] and run many environments in parallel, each of which has actors with varying exploration rates that add to a centralized replay buffer.

The simplest way to adapt deep Q-learning to the Dec-POMDP setting is through Independent Q-learning (IQL) as proposed by [32]. In IQL, every agent individually estimates the total return and treats other agents as part of the environment. There are other methods that explicitly account for the multi-agent structure by taking advantage of the *centralized training with decentralized control* regime [31, 28]. However, since our work is based on learning *best responses*, here we only consider IQL.

## 3.3 Zero-Shot Coordination Setting

Generally, many past works have focused on solving solving the *self-play* (SP) case for Dec-POMDPs. However, as shown in [18], these policies typically lead to arbitrary handshakes that work well within a team when jointly training agents together, but fail when evaluated with other independently trained agents from the same algorithm or humans. However, many real-world problems require interaction with never before seen AI agents and humans.

This desiderata was formalized as the *zero-shot coordination* (ZSC) by [18], in which the goal is to develop algorithms that allow *independently trained* agents to coordinate at test time. ZSC requires agents not to rely on *arbitrary* conventions as they lead to mutually incompatible policies across different training runs and implementations of the same algorithm. While extended episodes allow for agents to adapt to each other, this must happen at test time *within* the episode. Crucially, the ZSC setting is a stepping stone towards human-AI coordination, since it aims to uncover the *fundamental principles* underlying coordination in complex, partially observable, fully cooperative settings.

Lastly, the ZSC setting addresses some of the shortcomings of the ad-hoc team play [30] problem setting, where the goal is to do well when paired with *any* well performing SP policy at test time. As Hanabi shows, this fails in settings where there is little overlap between good SP policies and those that are suitable for coordination. So notably in our *ad-hoc* experiments we do not use SP policies but instead ones that can be meaningfully coordinated with.

## 4 Cognitive Hierarchies for ZSC

The methods we investigate and improve upon in this work are multi-agent RL adaptations of behavioral game theory's [3] cognitive hierarchies, where level $k$ agents are a BR to all preceding levels $\{0, \cdots k-1\}$; we define CH's as a Poisson distributions over all previously trained levels. We consider *k-level reasoning* (KLR) to be a hierarchy wehre level $k$ agents are trained as an approximate BR to level $k-1$ agents [7]. Lastly, we propose a new type of hierarchy, SyKLRBR, which is a hybrid of the two, where we train a BR to a Poisson distribution over all levels of a KLR (see appendix A.1 for more details).

---

**Algorithm 1:** Client-Server Implementation of k-level reasoning, cognitive hierarchies, SyKLRBR.

---

1: **Inputs**: a level $k$
2: **Initialization:** From the server retrieve a trainable policy $\pi_k$ and corresponding set of collaborative policies
    $\Pi_k$, $\Pi_k = \{\pi_{k-1}\}$ for k-level reasoning, $\Pi_k = \{\pi_0, \cdots, \pi_{k-1}\}$ for cognitive hierarchies, and
    $\Pi_{BR.} = \{\pi_0, \cdots, \pi_k\}$ for the Best Response Agent in SyKLRBR.
3: iteration = 0;
4: **for** epoch in $\{1, \cdots \text{num\_epochs}\}$ **do**
5:   **for** iter in $\{1, \cdots \text{num\_iter\_per\_epoch}\}$ **do**
6:     **if** iter $\%$ server\_update $== 0$ **then**
7:       UpdateWeightsOnServer($\pi_k$)
8:       RetrieveServerWeights($\Pi_k$)
9:     **end if**
10:     Update weights for $\pi_k$ towards a best response to $\Pi_k$
11:   **end for**
12: **end for**

---

For all hierarchies, we start training the first level of the hierarchy $\pi_1$ as an approximate BR to a uniform random policy over all legal actions [2], $\pi_0$. The main idea of this choice is that it prevents the $\pi_0$ agent from communicating any information through its actions, beyond the *grounded information* revealed by the environment (see [17] for more info). It thus forces the $\pi_1$ agent to only play based on this grounded information provided, without any conventions.

Furthermore, it is a natural choice for solving *zero-shot coordination* problems since it makes the least assumptions about a specific policy and certainly does not break any potential symmetries in the environment. Crucially, as is shown in [3], in *principle*, the convergence point of CH and KLR should be a deterministic function of $\pi_0$ and thus a common-knowledge $\pi_0$ should allow for zero-shot coordination between two independently trained agents.

A typical implementation of these training schemas is to train all levels sequentially, one level at a time, until the given level has converged. We also draw inspiration from [23] and their deep cognitive hierarchies framework (DCH) to instead train all levels simultaneously. To do so, we use a central server to store the policies of a given hierarchy and periodically refresh these policies by sending updated policies to the server and retrieving policies we are best responding to from the central server.

We implement the *sequential* training as follows: We halt the training of a policy $\pi_k$ at a given level $k$ after 24 hours and start training the next level $\pi_{k+1}$ as a BR to the trained set of policies $\Pi_k$, where $\Pi_k = \{\pi_0, \cdots, \pi_{k-1}\}$ for the CH case and $\Pi_k = \{\pi_{k-1}\}$ in the KLR case. This is the standard implementation of KLR and CH, as it was unsuccessfully explored in Hanabi by [18].

For *synchronous* training we train all levels in parallel under a client-server implementation (see algorithm 1). Here all policies $\{\pi_1, \cdots, \pi_n\}$ are initialized randomly on the server. A client training a given level $k \in \{1, \cdots, n\}$, fetches a policy $\pi_k$ and corresponding set of partner policies $\Pi_k$ and trains $\pi_k$ as an approximate BR to $\Pi_k$. Periodically, the client sends a copy of its updated policy $\pi_k$, fetches an updated $\Pi_k$ and then continues to train $\pi_k$. The entire hierarchy is synchronously trained for 24 hours, the same amount as a single level is trained in the sequential case.

## 5 Experimental Setup

### 5.1 Hanabi Setup

Hanabi is a cooperative card game that has has been established as a complex benchmark for fully cooperative partially observable multi-agent decision making [1]. In Hanabi, each player can see every player's hand but their own. As a result, players can receive information about their hand either by receiving direct (grounded) "hints" from other players, or by doing *counterfactual reasoning* to interpret other player's actions. In 5-card Hanabi, the variant considered in this paper, there are 5 colors (**G**, **B**, **W**, **Y**, **R**) and 5 ranks (**1** to **5**). A "hint" can be of color or rank and will reveal all cards of the underlying color or rank in the target player; an example hint is, "your first and fourth cards

---

[2] In Hanabi there are some illegal moves, *e.g.*, an agent cannot provide a hint when the given color or rank is not present in the hand of the team mate.

are **1**s." A caveat is that each hint costs a scarce *information* token, which can only be recovered by "discard" a card.

The goal in Hanabi is to complete 5 stacks, one for each of the 5 colors, each stack starting with the "1" and ending with the "5". At one point per card the maximum score is 25. To add to a stack players "play" cards and cards played out of order cost a *life token*. Once the deck is exhausted or the team loses all 3 lives ("bombs out"), the game will terminate.

## 5.2 Training Details

For a single agent we utilize distributed deep recurrent Q-Networks with prioritized replay experience [21]. Thus, during training there are a large number of simultaneously running environments calling deep Q-networks to generating and adding trajectories to a centralized replay buffer, which are then used to update the model. The network calls are dynamically batched in order to run efficiently on GPUs [11]. This agent training schema for Hanabi was first used in [15], and achieved strong results in the self-play setting. Please see the Appendix A for complete training details.

## 5.3 Evaluation

We evaluate our method and baseline in both self-play (SP), zero-shot coordination (ZSC), ad-hoc teamplay and human-AI settings. For zero-shot coordination, we follow the problem definition from [18] and evaluate models through cross-play (XP) where we repeat training 5 times with different seeds and pair the *independently* trained agents with each other.

To test our models' performance against a diverse set of unseen, novel partners (*ad-hoc* team play [30]), we next use RL to train two types of agents that use distinct conventions. The first RL agent is trained with Other-Play, which almost always hints for the rank of the playable card to communicate with their partners. For example, in a case where "Red 1" and "Red 2" have been played and the partner just draw a new "Red 3", the other agent will hint 3 and then partner will play that card deeming that 3 being a red card based on the convention. This agent is therefore referred to as ***Rank Bot***. The second RL agent is a color-based equivalent of Rank Bot produced by adding extra reward for color hinting during early stage of the training to encourage color hinting behavior. This agent is called ***Color Bot***. More details are in the appendix.

We also train a supervised bot (***Clone Bot***) on human data, as a proxy evaluation for zero-shot human-AI coordination. We used a dataset of $208,974$ games obtained from Board Game Arena (https://en.boardgamearena.com/). During training, we duplicate the games so that there is a training example from the perspective of each player, for a total of $417,948$ examples; that is, observations contain visible private information for exactly one of the players (the other being hidden). Using this dataset, we trained an agent to reproduce human moves by applying behavioral cloning. The agent is trained by minimizing cross-entropy loss on the actions of the visible player. After each epoch, the agent performs 1000 games of self-play, and we keep the model with the highest self-play score across all epochs.

| Level | Self-play | Cross-Play | w/ (k-1)th level | XP (k-1)th level | w/ Color Bot | w/ Rank Bot | w/ Clone Bot |
|---|---|---|---|---|---|---|---|
| 1 | $3.64 \pm 0.50$ | $3.82 \pm 0.19$ | $0.03 \pm 0.00$ | $0.03 \pm 0.00$ | $2.94 \pm 0.49$ | $3.84 \pm 0.33$ | $3.03 \pm 0.37$ |
| 2 | $10.36 \pm 1.14$ | $10.08 \pm 0.55$ | $6.66 \pm 1.35$ | $7.15 \pm 0.45$ | $9.49 \pm 0.98$ | $7.79 \pm 0.66$ | $7.64 \pm 1.12$ |
| 3 | $13.32 \pm 1.29$ | $13.10 \pm 0.67$ | $17.99 \pm 0.21$ | $12.62 \pm 0.95$ | $10.49 \pm 0.94$ | $8.03 \pm 1.13$ | $8.80 \pm 1.26$ |
| 4 | $15.63 \pm 2.35$ | $14.18 \pm 1.31$ | $21.41 \pm 0.15$ | $15.48 \pm 1.33$ | $13.63 \pm 2.09$ | $\mathbf{12.47 \pm 0.79}$ | $12.33 \pm 1.90$ |
| 5 | $\mathbf{16.97 \pm 1.19}$ | $\mathbf{17.17 \pm 0.98}$ | $\mathbf{22.77 \pm 0.08}$ | $17.04 \pm 1.54$ | $\mathbf{14.80 \pm 1.77}$ | $12.36 \pm 1.44$ | $\mathbf{13.03 \pm 1.91}$ |

Table 1: Performance of sequentially trained KLR for *Self-play* (SP), *Cross-Play* (XP), with the $k-1$th level, XP with the $k-1$th level, with Color Bot, with Rank Bot, and with Clone Bot. We find that the score with level $k-1$ drops from over 22 points to roughly 17 in XP. This indicates that in the sequential training, each level-$k$ can *overfit* to the static level-$k-1$ and thus develop arbitrary handshakes that propagate along the hierarchy.

## 6 Results and Discussion

In this section we present the main results and analysis of our work, for *sequential training*, *synchronous training*, and *SyKLRBR*. For each variant/level we present *self-play*, *cross-play*, *ad-hoc*

| Level | Self-play | Cross-Play | w/ (k-1)th level | XP (k-1)th level | w/ Color Bot | w/ Rank Bot | w/ Clone Bot |
|---|---|---|---|---|---|---|---|
| 1 | $3.95 \pm 0.28$ | $4.55 \pm 0.11$ | $0.03 \pm 0.00$ | $0.03 \pm 0.00$ | $3.44 \pm 0.32$ | $4.59 \pm 0.22$ | $3.70 \pm 0.28$ |
| 2 | $14.97 \pm 0.31$ | $15.67 \pm 0.09$ | $9.95 \pm 0.29$ | $9.81 \pm 0.12$ | $8.40 \pm 0.11$ | $7.71 \pm 0.25$ | $9.95 \pm 0.54$ |
| 3 | $20.00 \pm 0.25$ | $20.67 \pm 0.08$ | $19.27 \pm 0.42$ | $18.85 \pm 0.21$ | $14.10 \pm 0.47$ | $14.64 \pm 0.36$ | $13.38 \pm 0.51$ |
| 4 | $21.28 \pm 0.22$ | $21.05 \pm 0.16$ | $22.61 \pm 0.12$ | $22.45 \pm 0.07$ | $15.80 \pm 0.57$ | $15.43 \pm 0.64$ | $14.09 \pm 0.18$ |
| 5 | $\mathbf{22.29 \pm 0.08}$ | $\mathbf{22.14 \pm 0.12}$ | $\mathbf{23.47 \pm 0.10}$ | $\mathbf{23.16 \pm 0.16}$ | $\mathbf{17.21 \pm 0.86}$ | $\mathbf{15.90 \pm 0.30}$ | $\mathbf{15.86 \pm 0.23}$ |

Table 2: Synchronously trained KLR performance for *Self-play* (SP), *Cross-Play* (XP), with the $k-1$th level, XP with the $k-1$th level, with Color Bot, with Rank Bot, and with Clone Bot. Synchronous training produces extremely *stable* outcomes across the different runs, as indicated by the close correspondence between SP and XP scores. The fact that all levels are *changing* during training regularizes the process and prevents overfitting to the level $k-1$.

*teamplay* and *human-proxy* results. Although we present self-play numbers, the purpose of this paper is not to produce good self-play scores, rather we are optimizing for the ZSC and ad-hoc settings. Therefore, our analysis focuses on the *cross-play* and *ad-hoc teamplay* settings, including the *human-proxy* results. We demonstrate that simply training the KLR synchronously achieves significant improvement over its sequentially trained counterpart in the ZSC setting. We also demonstrate that our new method SyKLRBR is able to further improve upon the synchronous KLR results and achieve SOTA results in certain metrics *e.g.* scores with clone bot. We also provide analysis into the issues with sequential training and how *synchronous* training addresses them.

## 6.1 XP Performance

Table 3 shows the XP scores for other-play, OBL, sequential KLR, synchronous KLR, and SyKLRBR. Changing the training schema from sequential to synchronous significantly increases the XP score to the *state-of-the-art* XP score for methods that don't use access to the environment or known symmetries. Thus, by synchronously training the KLR, we are able to achieve strong results in the ZSC setting without requiring underlying game symmetries (other-play) or using simulator access (OBL). SyKLRBR improves upon this result by synchronously training the BR and the KLR, yielding even better XP results. Additionally, tables 1 and 2 show the performance of all levels of KLR. A KLR trained sequentially or synchronously is able to achieve good scores with the $k-1$th level, as level $k$ is explicitly optimized to be an approximate best response to level $k-1$. However, the sequential KLR has a significant dropoff for the XP score with the $k-1$th level, indicating that sequential KLRs have large inconsistencies across runs. This also indicates that the sequentially trained hierarchy is overfitting to the exact $k-1$th level. In contrast, the synchronously trained hierarchy keeps its score with the $k-1$th level close to the XP score with the $k-1$ level. Thus, by synchronously training the hierarchy we are able to minimize overfitting. For more analysis on overfitting see section 6.4.

## 6.2 Ad-hoc Teamplay

Table 3 shows the scores in the ad-hoc teamplay setting i.e. evaluation with color and rank bot, where the synchronously trained KLR outperforms the sequentially trained KLR for both bots. Similarly, our SyKLRBR further improves performance with both rank and color bot. Thus, the benefits from synchronous training and from training a BR measured in the ZSC setting translate to improvements in ad-hoc teamplay.

## 6.3 Zero-Shot Coordination with Human Proxies

Up to now we have focused on AI agents playing well with each other. Next we measure performance of bots playing with bots trained on human data, representing a human proxy, specifically the Clone Bot described in Section 5.3. In table 3, we present overall performance of our agents when trained under OP, OBL, sequentially KLR, synchronously KLR, and SyKLRBR. As a reference, we trained a bot using [18]'s OP, which when paired with Clone Bot achieved an average score score of $8.55\pm0.48$.

When synchronously trained, KLR monotonically improves its score with Clone Bot. By level 5 the synchronously trained KLR is able to achieve a score of $15.801 \pm 0.26$; the sequentially trained KLR has a significantly lower score. Additionally, the synchronously trained KLR Clone Bot score is comparable to the more algorithmically complex OBL bot, which furthermore requires access

| Method | Self-play | Cross-Play | w/ Color Bot | w/ Rank Bot | w/ Clone Bot | Limitations |
|---|---|---|---|---|---|---|
| Other-Play | **24.14 ± 0.03** | 21.77 ± 0.68 | 4.05 ± 0.37 | - | 8.55 ± 0.48 | Sym |
| OBL (level 4) | 24.10 ± 0.01 | **23.76 ± 0.06** | **21.78 ± 0.42** | 14.46 ± 0.59 | 16.00 ± 0.13 | Env |
| Sequential | 16.97 ± 1.19 | 17.17 ± 0.98 | 14.80 ± 1.77 | 12.36 ± 1.44 | 13.03 ± 1.91 | - |
| Synchronous | 22.29 ± 0.08 | 22.14 ± 0.12 | 17.21 ± 0.86 | 15.90 ± 0.30 | 15.86 ± 0.23 | - |
| SyKLRBR | 23.40 ± 0.07 | 23.29 ± 0.05 | 17.62 ± 0.69 | **17.01 ± 0.45** | **16.59 ± 0.16** | - |

Table 3: Other-Play, OBL level 4, level 5 Sequential KLR, level 5 Synchronous KLR, and SyKLRBR for *Self-play* (SP), *Cross-Play* (XP), ad-hoc play with Color Bot, Rank Bot, and Clone Bot. We include methodological limitations, requiring underlying game symmetries (sym) or requiring access to the simulator (env). Both synchronous training and our SyKLRBR improve upon XP, ad-hoc teamplay, and clone-bot scores. Also SyKLRBR achieves state-of-the-art results with clone-bot.

to the simulator. Lastly, our new method, SyKLRBR, is able to achieve state-of-the-art results in coordination with human proxies. Therefore, through simply synchronously training KLR we are able to produce bots that cooperate well with human-like proxy policies at test time and by co-training a BR we obtain state-of-the-art results.

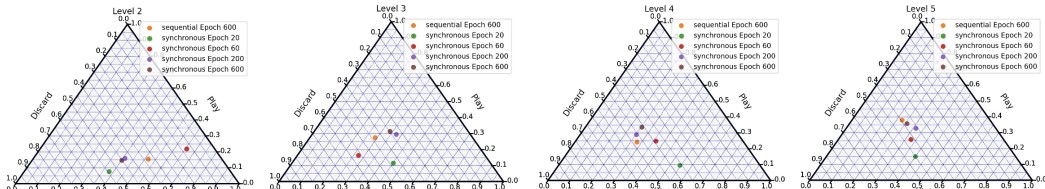

Figure 2: A plot of probability distributions of actions an agent at level $k - 1$ will play with level $k$ for KLR trained sequentially and synchronously. At lower levels, the synchronous KLR is stochastic, but at higher levels it stabilizes. The stochasticity in the lower levels broadens the range of policies seen and robustifies lower levels, which propagates upwards and leads to stable, robust policies.

| Training Schema | % Bomb out with (k-1)th level | % Bomb out in XP with (k-1)th level |
|---|---|---|
| Sequential | 0.4 ± 0.2 | 19.8 ± 18.0 |
| Synchronous | 0.4 ± 0.2 | 0.6 ± 0.3 |

Table 4: Percentage of bombing out for the level 5 agent playing with the level 4 agent it trained with ((k-1)th level) or the other level 4 agents unseen at training time (XP with (k-1)th level). By synchronously training we prevent overfitting to the policy distribution of a fixed agent. This allows us to be better off distribution, which significantly reduces bombing out in the XP case.

### 6.4 Observations of Training Behaviors

We plot the probability that an agent from $k - 1$ will take a given type of action $u$ when playing with an agent from level $k$ in Figure 2. At low levels of the hierarchy (levels 2, 3), the synchronous hierarchy is trained as an approximate best response to a set of changing policies. Higher up in the hierarchy the change in the policies gets attenuated, leading to stable policies towards the end of training. By synchronously training the hierarchy, we allow each policy to see a wider distribution of states and ensure it is robust to different policies at test time. This robustness is reflected in the improved ZSC, XP with $k - 1$th levels, ad-hoc teamplay, and human-proxy coordination.

In table 4 we present the percentage of "bombing out" for the level 5 agent playing with the level 4 agent it trained with or level 4 agents from other seeds of our KLRs. "Bombing out" is a failure case when too many unplayable cards have been played, leading to the agent losing all points in the game. Both the sequential and synchronous KLRs rarely bomb out when paired with their training partners. Only the sequential KLR bombs out significantly more in XP, roughly 20% compared with <1% with the agent it trained with. This high rate illustrates that the agent is making a large number of mistakes, indicating that it is off-distribution in XP. We verified this by checking the Q-values of the action the agent takes when it bombs out. The vast majority of cases (90%+) the agent has a positive Q-value

for its *play action* when it bombs out and negative Q-values associated with other actions (discarding and hinting). Since the play action is causing the large negative reward, while the other actions are safe, these Q-values are clearly nonsensical, another indicator that the agent is off-distribution. All of this illustrates that the "bomb out" rate is a good proxy for being off-distribution, which shows that the synchronously trained KLR agents are more on-distribution during XP testing.

### 6.5 Understanding Synchronous Training

At a training step $t$, the synchronous KLR $\pi_i^t$ is trained towards a BR to $\pi_{i-1}^t$. There are a few reasons why synchronous training helps regularize training. First of all, weights are typically initialized s.t. Q-values at the beginning of training are small, so under a softmax all $\pi_i^0 \ \forall i \in k$ are close to uniform. Secondly, over the course of training the entropy for each policy decreases, as Q-values become more accurate and drift apart, so $\pi_i^T$ (the final policy) will have the lowest entropy. Lastly, the entropy of the average policy across the set $\{\pi_{i-1}^1, \pi_{i-1}^2, \cdots, \pi_{i-1}^T\}$ is higher than the average of the entropies from the same set (e.g. the average of two deterministic policies is stochastic, but the average entropy of the policies 0). Therefore, by playing against a changing distribution over stochastic policies we significantly *broaden* the support of our policy.

Entropy in $\pi_{i-1}$ has two effects: First of all it increases robustness by exposing $\pi_i$ to more states during training and, secondly, more entropic (i.e. random) policies will generally induce *less informative* posterior (counterfactual) beliefs (a *fully random* policy is the extreme case, with a completely uninformative posterior). As a consequence, the BR to a set of different snapshots of a policy $\pi_{i-1}^t$ is both more robust and less able to extract information from the actions of $\pi_{i-1}$ than the BR to only the final $\pi_{i-1}^T$. This forces the policy to rely to a greater extend on *grounded information* in the game, rather than arbitrary conventions.

Empirically we show this effect by training a belief model on a $\pi_1^T$ and on a set of snapshots of $\pi_1^t$ for $t = (100, 200, ...1000)$. The cross entropy of belief model for the final $\pi_1^T$ is $1.58 \pm 0.01$, while the cross entropy for the set is substantially higher ($1.70 \pm 0.01$) (both averaged over 3 seeds).

### 6.6 Cognitive Hierarchies (CH)

We also use our synchronous setup to train a CH (*i.e.*, a best response to a Poisson sample of lower levels) and present the results in table 5. We note that the scores for the synchronous CH are lower than the synchronous KLR in terms of SP, XP, ad-hoc teamplay, and human-proxy coordination. This is likely because even at higher levels, the majority of the partner agents come from lower levels, as a result the performance is similar to that of KLR level 3. Additionally, computing a best response to a mix of lower level agents makes the hints provided less reliable and disincentivizes the agent to hint.

| Level | Self-play | Cross-Play | w/ Color Bot | w/ Rank Bot | w/ Clone Bot |
|---|---|---|---|---|---|
| 1 | $4.11 \pm 0.29$ | $4.33 \pm 0.12$ | $4.06 \pm 0.30$ | $4.72 \pm 0.40$ | $4.21 \pm 0.28$ |
| 2 | $15.61 \pm 0.27$ | $15.79 \pm 0.10$ | $8.77 \pm 0.61$ | $8.58 \pm 0.83$ | $9.26 \pm 0.51$ |
| 3 | $19.71 \pm 0.12$ | $19.51 \pm 0.07$ | $10.43 \pm 0.29$ | $11.52 \pm 0.63$ | $13.10 \pm 0.26$ |
| 4 | $20.85 \pm 0.17$ | $20.57 \pm 0.08$ | $12.90 \pm 0.42$ | $13.75 \pm 0.47$ | $13.59 \pm 0.33$ |
| 5 | $\mathbf{21.65 \pm 0.21}$ | $\mathbf{21.42 \pm 0.10}$ | $\mathbf{14.13 \pm 0.16}$ | $\mathbf{15.12 \pm 0.27}$ | $\mathbf{14.59 \pm 0.25}$ |

Table 5: CH synchronously trained performance for *Self-play* (SP), *Cross-Play* (XP), and ad-hoc play with Color Bot, Rank Bot, and Clone Bot. In the CH setting we are unable to obtain very strong results, regardless of setting.

## 7 Conclusion

How to coordinate with independently trained agents is one of the great challenges of multi-agent learning. In this paper we show that a classical method from the behavioral game-theory literature, *k*-level reasoning, can easily be adapted to the deep MARL setting to obtain strong performance on the challenging benchmark task of two player Hanabi. Crucially, we showed that a simple engineering decision, to train the different levels of the hierarchy at the same time, made a large difference for the final performance of the method. Our analysis shows that this difference is due to the *changing*

policies at lower levels regularizing the training process, preventing the higher levels from overfitting to specific policies. We have also developed a new method SyKLRBR, which further improves on our synchronous training schema and achieves state-of-the-art results for ad-hoc teamplay performance.

This raises a number of possibilities for follow-up work: What other ideas have been unsuccessfully tried and abandoned too early? Where else can we use *synchronous training* as a regularizer? Another interesting avenue is to investigate whether the different levels of the hierarchy are evaluated *off-distribution* during training and how this can be addressed. Level-$k$ is only trained on the distribution induced when paired with level-$k - 1$, but evaluated on the distribution induced from playing with $k + 1$. Furthermore, extending the work of [13] and searching for the optimal graph-structure during training is a promising avenue for future work.

## 8 Limitations

Although our synchronous training schema does alleviate overfitting in the KLR case, there is still a large gap between cross-play and playing with the $k - 1$th level. This indicates that there still exist some unfavorable dynamics in the hierarchy. Similarly, although our work does provide steps towards human-AI cooperation, the policy can still be brittle with unseen bots resulting in lower scores.

## 9 Broader Impact

We have demonstrated that synchronously training a KLR greatly improves on sequentially training a KLR in the complex Dec-POMDP setting, Hanabi. This in essence is a simple engineering decision, but it improves performance to very competitive methods. Our method, SyKLRBR, synchronously trains a BR to the KLR, which resulted in SOTA performance for coordination with human proxies through "clone bot." We have found that our method works as it provides distributional robustness in the trained policies. As a result, it can be a positive step towards improving human-AI cooperation. Clearly no technology is safe from being used for malicious purposes, which also applies to our research. However, fully-cooperative settings are clearly targeting benevolent applications.

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
