| Hyper-parameters | Value |
|---|---|
| *Replay Buffer Parameters* | |
| burn-in-frames | 10000 |
| replay buffer size | 131072 ($2^{17}$) |
| priority exponent | 0.9 |
| priority weight | 0.6 |
| maximum trajectory length | 80 |
| *Optimization Parameters* | |
| optimizer | Adam [22] |
| lr | $6.25e-05$ |
| eps | $1.5e-5$ |
| gradient clip | 5 |
| batchsize | 128 |
| *Q-learning Parameters* | |
| n step | 3 |
| discount factor | 0.999 |
| num gradient steps sync target net | 2500 |

Table 6: Hyper-parameters for Hanabi agent training

## A   Experimental Details

In training every agent we use a distributed framework for simulation and training. For simulation, we run 6400 Hanabi environments in parallel and the trajectories are batched together for efficient GPU computation. This is done efficiently as every thread can hold many environments in which many agents interact. Every agent chooses actions based on neural network calls, which are more intensive and done by GPUs. By doing these calls asynchronously it allows a thread to support multiple environments while waiting for prior agents' actions to be computed. Therefore, by stacking multiple environments into a thread and utilizing multiple threads we are able to maximize GPU utility and generate a massive amount of data on the simulation side. Every environment is considered to be in a permanent simulation loop, where at the end of the environment the entire action observation history, consisting of action, observation, and reward is aggregated together into a trajectory, padded to a length 80, and then added to a centralized replay buffer as done in [29]. We compute the priority of each trajectory as $\xi = 0.9 \cdot max_i \xi_i + 0.1 \cdot \xi$ [21], where $\xi_i$ is the TD error per step. From the training perspective we have a training loop that continuously samples trajectories from the replay buffer and updates the model based on TD error. The simulation policies are updated to be the training policy every 10 gradient steps. We utilize epsilon exploration for training agent exploration. At the beginning of every simulated game we generate epsilon $\epsilon_i$ from the equation $\epsilon_i = \alpha^{1+\beta*\frac{i}{N-1}}$, where $\alpha = 0.1, \beta = 7, N = 80$. For our entire training, inference infrastructure we use a machine with 30 CPU cores and 2 GPUs, one GPU for training and one GPU for simulation.

We use the same network architecture as described in [16]. We follow their design choices of utilizing a 3-layer feedforward neural network to encode the entire observation and then using a one-layer feedforward neural network followed by an LSTM to encode only the public observation. We combine these two outputs with element-wise multiplication and use a dueling architecture [36] to get the final Q-values. We also use double DQN as done in [34]. Other relevant hyper-parameters are presented in table 6.

For synchronous hierarchy training, every 50 gradient steps, each client sends the weights of the policy it is training $\pi_i$ to the server and queries the server for the corresponding set of updated policies $\Pi_i$ that $\pi_i$ is trained to be an approximate best response.

### A.1   Poisson Distribution Details

For CH and SyKLRBR, each responds to a Poisson distribution over some set of agents $\{\pi_0, \pi_1, \cdots \pi_k\}$. Concretely, each of the games played simultaneously has an agent from a set level. We use a Poisson distribution with a PMF of $\frac{\lambda^k * e^{-\lambda}}{k!}$. For SyKLRBR we use $\lambda = 1$, which means for a given level $j$ and a hierarchy of $i$ levels $k = i - j$ in the PMF. Therefore, a BR to a 5

level KLR has $\sim 37\%$ of the actors from level 5, $\sim 37\%$ from level 4, $\sim 18\%$ from level 3, $\sim 6\%$ from level 2, $\sim 1\%$ from level 1, and $< 1\%$ from level 0.

Similarly, for CH we use $\lambda = 2$, which is a standard value for CHs as noted by [4]. Thus, a CH at a given level $i$ and partner level $j$, it will have $k = j$ in the Poisson PMF for a given level $j$ (excluding level 0). Therefore, for a 5 level cognitive hierarchy, $\sim 37\%$ of the actors are from level 1, $\sim 37\%$ from level 2, $\sim 20\%$ from level 3, and $\sim 6\%$ are from level 4.

## B    Details on Rank Bot and Color Bot

We train two distinct policies to test the ad-hoc teamplay performance of our agents. Both two policies use the same network design as our KLR policies. The first policy is trained with the Other-Play [18] technique where one of the two players always observe the world, i.e. both input observation and output action space, in a randomly permuted color space. The color permutation is sampled once at the beginning of each episode. This method is capable of preventing the agent from learning arbitrary conventions and previously achieved the best zero-shot coordination score in Hanabi. Empirically, policies trained with Other-Play tends to use a rank based convention where it hints about the rank of a playable card to indicate play and partner will often safely play a rank hinted card without knowing the color. Therefore we refer to this policy as ***Rank Bot***. Similarly, we may expect a color based equivalent of the Rank Bot but in practice we find it difficult to learn such policy naturally. We instead use a reward shaping technique where we give extra reward of 0.25 when the agent hints a color. To wash out the artifact of the reward shaping, we first train the agent with reward shaping till convergence and then disable the extra reward and train it for another 24 hours. However, we find that the reward shaping may lead to inconsistent training results across different runs and thus make it hard to reproduce. We use a simple trick of zeroing out the last action field of the observation to stabilize the learning. Note that the last action is a shortcut to learn arbitrary conventions but it is redundant in our setting since the agent with RNN can infer last action from the board. The policy trained this way predominantly uses color based conventions and is referred to as ***Color Bot***.