# OpenReview forum: "K-level Reasoning for Zero-Shot Coordination in Hanabi"
_NeurIPS.cc/2021/Conference — NeurIPS 2021 Poster_

### Official Review · Reviewer_4AuN · 2021-07-15

**Rating:** 5
**Confidence:** 2

**Summary:**

This paper proposed a new method to solve the Hanabi game.

**Limitations And Societal Impact:**

1. They should evaluate their method on more benchmarks, such as MPE, google football and other card games.

2. They should compare their method with recent MARL algorithms, such as MAPPO.

**Main Review:**

This paper utilized k-level reasoning to solve the Hanabi game. Their method is quite simple. But I think they should give more theoretical analysis and demonstrate the effectiveness of their method. For example, they should compare their method with MAPPO[1]. And they also need to demonstrate the generalization of their method by applying their method to other multi-agent games.



[1] Yu C, Velu A, Vinitsky E, et al. The surprising effectiveness of mappo in cooperative, multi-agent games[J]. arXiv preprint arXiv:2103.01955, 2021.

After reading the author's response, I have increased the rating from 3 to 5.

**Time Spent Reviewing:**

5

---

> ### Author Response · Authors · 2021-08-11
> **Reply to Reviewer 4AuN**
>
> We’d like to address the points brought up by the reviewer:
>
> **They should evaluate their method on more benchmarks, such as MPE, google football and other card games.**
>
> The goal of our paper is not to “solve the Hanabi game”, as stated by the reviewer. Instead, we are addressing the zero-shot coordination problem which assumes fully cooperative, partially observable settings. Unfortunately, the environments listed are either not fully cooperative (Google football, most card games) and/or have not been evaluated from a coordination perspective (MPE). Introducing new testbeds properly is a major amount of work and thus, while useful, outside the scope of this paper.
>
> **They should compare their method with recent MARL algorithms, such as MAPPO.**
>
> It is well known that self-play algorithms, such as variations of Q-learning (e.g. VDN, QMIX etc) or actor-critic methods (e.g. MAPPO) do poorly in ad-hoc teamwork and Zero-shot coordination in Hanabi. However, they can be used for the ‘inner-loop’ of our method. For simplicity we currently use IQL, but the actual learning algorithm used is entirely orthogonal to our contributions and we leave it to future work to explore this further.

---

> > ### Author Response · Authors · 2021-08-18
> > **Following up with Reviewer 4AuN**
> >
> > We thank you for taking the time to review. We wanted to follow up with the reviewer to see if they had any more questions/feedback. Additionally, as a summary of our initial response:
> >
> > 1. The goal of the paper wasn't to "solve the Hanabi game." Additionally, the provided environments are either not fully cooperative (Google football, most card games) and/or haven't been evaluated from a coordination perspective (MPE).
> > 2. It is well known that self-play algorithms do poorly in ad-hoc teamwork and zero-shot coordination in Hanabi. However, MAPPO could be used as part of the 'inner-loop' for our method, where as of now we use IQL.
> >
> > Please let us know if you have any more questions.
> >
> > Best,
> >
> > Authors of Paper10963

---

> > ### Comment · Reviewer_4AuN · 2021-08-18
> > **Thanks for your response.**
> >
> >  > Why do I ask for more games for evaluation?
> >
> > 1. In your response, you state that your goal is not to solve Hanabi. However, you place `Hanabi` in your title which means that your paper aims to develop a method to solve a specific task. Why don't you use `K-level Reasoning for Zero-Shot Coordination Problems` as your title?
> > 2. You state that your goal is to solve the zero-shot coordination problem. So I think you should evaluate your method on various tasks to demonstrate the effectiveness and generality of your method on such problems. But you only test your method on Hanabi.
> >
> > > Why do I ask you to compare with MAPPO?
> >
> > MAPPO(https://arxiv.org/abs/2103.01955) has been evaluated on Hanabi. I think it is critical for you to compare your method with the current deep RL method. If you can achieve significant improvement compared to MAPPO, I think it will greatly enhance the value of your paper.
> >
> > I will stick to the current rating unless you can give out convincing experimental results.

---

> > > ### Author Response · Authors · 2021-08-19
> > > **Response / reiteration**
> > >
> > > @1: We can update the title to clarify that while we are addressing ZSC in general, we only evaluate on Hanabi (currently the standard testbed for ZSC).
> > > @2: We addressed this in a fair amount of detail in our response, but I'll repeat it here. Benchmarking and developing new environments for ZSC is a good idea but, when done properly, in itself a fair amount of work and thus beyond the scope of this paper. Other-play and OBL are both accepted at top conferences and only evaluated on two-player Hanabi. An interesting option might be to run on e.g. 3 player Hanabi.
> > >
> > > @MAPPO: MAPPO has not been evaluated in ZSC. However, other self-play algorithms have been and, as expected, they do extremely poorly (e.g. VDN, Impala, SAD). So while we will try to run the evaluation for MAPPO in ZSC, it's not a very meaningful number - the algorithm is simply not designed for this setting. I'll repeat that beating MAPPO in *self-play* is *not* a goal of this work.
> > >
> > > Thank you for your fast response and please let us know any further comments/questions.

---

> > > ### Author Response · Authors · 2021-08-27
> > > **MAPPO Evaluation Follow-Up**
> > >
> > > @MAPPO: We contacted the authors of the MAPPO paper to obtain the trained weights for 4 of their best agents. We then evaluated these in self-play and cross-play (both averaged over those 4 seeds):
> > >
> > > Self-play: $24.02 \pm 0.12$
> > >
> > > Cross-play: $15.55 \pm 3.81$
> > >
> > > While the Cross-play score is a lot lower than the $>23$ obtained by SyKLRBR it is much higher than the ~3 points obtained by other self-play methods, e.g. SAD in the OP paper. One explanation is that these MAPPO agents are feedforward, which drastically reduces the policy space. Higher XP scores for feedforward agents were also observed in the Hanabi challenge paper. However, ignoring history is not a general method and the somewhat reasonable XP score is an accidental occurrence, rather than a property of the method.
> > >
> > > Additionally, we plotted the conditional action matrices as done in Hu et al. 2020 (“Other-Play” for Zero-Shot Coordination ICML 2020). The best seed of MAPPO resulted in a highly arbitrary convention, discarding the 2nd card to signal that the first card should be played. This kind of failure is expected as MAPPO isn’t designed for zero-shot coordination. However, to reiterate, we could use MAPPO for the inner loop of our training algorithm.
> > >
> > > Please let us know if there are any further follow-up questions.

---

> ### Author Response · Authors · 2021-08-27
> **Thank you, Follow-up Requested**
>
> Given we're nearing the end of the discussion period, we wanted to follow up with the reviewer again.
>
> We’d like to thank the reviewer for their continued discussion and for raising their score. We also wanted to ask the reviewer if they could provide a rationale for this score over a higher one and ask if there’d be anything else they’d want to see to raise their score.

---

### Official Review · Reviewer_NyYA · 2021-07-16

**Rating:** 6
**Confidence:** 3

**Summary:**

This paper demonstrates an approach to hierarchically train agents for Hanabi, adapted from a classical approach called k-level reasoning. The authors demonstrate that training these hierarchies in parallel improves performance by reducing overfitting. They present a novel training schema they call SyKLRBR, which demonstrates SOTA performance with a couple of agents. The paper ends with a large number of experiments and discussion.

**Limitations And Societal Impact:**

The major limitation I think relates to the discussion above, that the authors have not actually evaluated their method alongside humans. However, I don't believe this is strictly necessary for the majority of their claims. In terms of social impact I don't foresee this being an issue for this particular paper.

**Main Review:**

Originality: The approach is certainly novel, both in terms of the simultaneous training and the specific SyKLRBR paradigm. The authors draw attention to the fact that they managed to get an abandoned older approach to work. However, I don't know the history of this approach sufficiently to comment on that.

Quality: The quality of the paper is quite high. The experiments are extensive and the discussion is rich. I have no major concerns when it comes to paper quality.

Clarity: The paper is overall fairly clear. However, there are a few issues. To start, Figure 1 is rather unclear. The visual language is inconsistent (why are some arrows on the left side of the screen and some on the right?), and many of the choices are not explained (the choice of colours and the upward arrows/floating dots). While I mostly picked up on what was going on from context, I think the figure might need to be revised substantially. Later, the authors claim "These datapoints... confirm that ZSC is a great proxy setting for human-AI coordination and ad-hoc team play". However, it is unclear what they mean by this. Further, the way that "clone bot" is referred to across the paper is unclear. Sometimes the clone bot is correctly referred to as a proxy, but at other times the authors make claims suggesting that performance with the clone bot is equivalent to improved performance with humans, such as "the performance gains directly translate into better human-AI coordination". This is unclear at best and overblown at worst.

Significance: The authors present a clear benefit to their adapted k-level reasoning. However, many of their claims (as mentioned in the last paragraph) are somewhat unclear. In particular, there's no evidence shown that clone bot is a particularly good proxy for human behaviour. The authors do not present any comparison between the human training data and clone bot to get a sense of how well it corresponds to human behaviour. Given that a significant portion of the author's claims rely on the quality of clone bot as a proxy, I would have expected something like that instead of some of the existing analysis (perhaps Table 4, which seems to just reinforce other results). However, even with this I think that the other results from the paper are significant enough to largely outweigh my concerns with clone bot.

**Time Spent Reviewing:**

4

---

> ### Author Response · Authors · 2021-08-11
> **Reply to Reviewer NyYA**
>
> We’d like to thank the reviewer for their detailed comments and helpful feedback. We address their comments below:
>
> We apologize for the clarity of figure 1 and we will revise it for the next version of the paper.
>
> **ZSC is a great proxy for ad-hoc teamplay and humanAI coordination**
>
> We were trying to express the fact that ZSC can serve as a proxy for ad-hoc teamplay and human-AI coordination, without requiring human data. From our experiments we demonstrated that as ZSC score increased it also typically led to an increase in ad-hoc teamplay scores and human AI coordination scores. Additionally, Off-Belief Learning Hu et al 2021, ICML 2021 similarly demonstrated for their OBL hierarchy increasing XP score led to better ad-hoc teamplay and clone-bot scores.
>
> **Clone bot quality concerns**
>
> Clone bot is indeed a proxy for human AI coordination. We agree that the claims can be strong  and we can tone down the language in the next paper iteration. However, clone bot should be considered a fairly accurate proxy for human performance; the scores of SAD and OP with clone bot closely match the human experiments in the “Other Play” paper by Hu et al. 2020. Additionally, there is also prior published work that utilizes clone bot as a human proxy e.g. Off-Belief Learning Hu et al 2021, ICML 2021. As a result, although clone-bot can only ever be a proxy for human performance, it serves as a fairly accurate model for human performance.

---

> > ### Author Response · Authors · 2021-08-18
> > **Following up with Reviewer NyYA**
> >
> > We again thank you for taking the time to review. We are following up to see if the reviewer had any more questions/feedback. Additionally, as a summary of our initial response:
> >
> > 1. We claim that ZSC is a great proxy for ad-hoc teamplay and humanAI coordination. In our work and "Off-Belief Learning" by Hu et al 2021 an increase in cross-play score led to better ad-hoc teamplay and clone-bot scores.
> > 2. Clone bot serves as an accurate proxy for human performance, as SAD and OP with clone bot closely match the reported results from human experiments in the "Other Play" paper by Hu et al. 2020. Additionally, prior work, "Off-Belief Learning" by Hu et al 2021, has used clone bot as a human proxy.
> >
> > Please let us know if you have any more questions.
> >
> > Best,
> >
> > Authors of Paper10963

---

> > > ### Comment · Reviewer_NyYA · 2021-08-25
> > > **Re: Following up with Reviewer NyYA**
> > >
> > > Hi Authors of Paper 10963,
> > >
> > > Thanks for your detailed comments, because of them I have higher confidence in Clonebot's appropriateness as a proxy for humans. However, this greater confidence did not lead to a change in rating. I still feel the paper should be accepted, but I think it could be made stronger with a new human subject study.
> > >
> > > Best wishes,
> > > Reviewer NyYA

---

### Official Review · Reviewer_VvuS · 2021-07-18

**Rating:** 7
**Confidence:** 4

**Summary:**

An important problem in multi-agent RL is how to train agents to cooperate and coordinate well with humans. One approach has been to train agents that can coordinate well with themselves, using self-play. Recently more attention has been paid to the problem of coordinating well with independent training runs: referred to as Zero-Shot Coordination, as an easier proxy for coordinating with particular other policies, such as humans. A benchmark game for this has been Hanabi, which is fully cooperative and, for strong play, requires sending signals whose meanings require multi-step chains of reasoning to deduce, resulting in complex conventions. One way to understand these chains has been via the so-called “cognitive hierarchy” (CH) from behavioural game theory, that was invented to characterize human non-equilibrium play in some settings: it’s a hierarchy in which higher levels select optimal strategies assuming other players are following strategies from lower levels.

Past work has tried to tackle ZSC in Hanabi using this idea directly (training a chain of policies that each best-respond to the preceding one), but the resulting training process did not manage to avoid arbitrary conventions, hurting ZSC performance. Recent work has explored more sophisticated ways of rooting out arbitrary conventions, eg by using carefully grounded priors and an expensive simulator to reason about the state distribution, or by ensuring that policies don’t learn to rely on a particular way of breaking symmetries in the game (via data augmentation).

This paper revisits the chain-of-best-responders method, and modifies the training schema by training each policy in the chain simultaneously. This results in policies that have near-SOTA ZSC performance; behavioural analysis is shown to support the explanation that this is because the moving-target opponents strongly regularize away the overfitting that otherwise occurs when training best-responders against fixed opponents. With another tweak (learning a best-response to a mixture of the chain of policies), a policy is trained that is even stronger in ZSC and self-play, and achieves SOTA in playing alongside a human-play proxy policy (as well as consistently good performance in some other ad-hoc teams). The presented method seems to be the first to achieve good ZSC in Hanabi without requiring bespoke components such as a model of all of the symmetries or a simulator to use for deriving belief distributions, and is therefore an important contribution that’s likely to be applicable to many other settings for learning coordination and strategic behaviour.

**Limitations And Societal Impact:**

The authors have adequately addressed the limitations and potential negative societal impact of their work.

**Main Review:**

This paper makes an important contribution to the literature by revealing that a previously-published negative result can be reversed to a strong positive result via some technical changes. The problem is well-motivated and relevant, and the paper is generally well-structured and informative.

The key method (synchronously trained KLR) is not very novel, nor is it accompanied by theoretical optimality results as other recent results in Zero-Shot Coordination have done. However, it is more readily applicable to more domains, the empirical results are competitive (and the comparison with previous methods is convincing), and the approach taken was dismissed as an unsuccessful idea in previous work.

The regularizing advantages of synchronous training are well-explored in the paper, with some interesting analyses of the effects on entropy, posteriors, and out-of-distribution-ness of bomb-outs. Another analysis that would be informative regarding the difference between sequential and synchronous training would be some measurement of the symmetry-invariance of the policy, since past work argued this was key to ZSC - this would also be interesting to compare with Other-Play, to see whether the XP benefit from synchronous training is accounted for.

But there are a couple of issues with other parts of the paper. For example, the paper presents an additional contribution SyKLRBR as an improvement on KLR, but since SyKLRBR requires training another BR alongside the K BR's in the KLR, the fair comparison would be with (K+1)LR. Extrapolating Table 2 suggests that KLR with K=6 may be on par with SyKLRBR with K=5; an experiment would be warranted to verify whether SyKLRBR provides any benefit.

Another thing that was unclear was the parameter of the Poisson distribution used in the implementation of both SyKLRBR and CH. The value wasn’t given, and eg a value $\rightarrow\infty$ would collapse both of them to KLR. (In the CH definition from Camerer et al, the Poisson distribution for level k needs to be restricted to 0, …, k-1 and rescaled.) As a result it’s unclear how to interpret the paper’s finding that CH significantly underperformed KLR.

**Time Spent Reviewing:**

6

---

> ### Author Response · Authors · 2021-08-11
> **Reply to Reviewer VvuS**
>
> We would like to thank the reviewer for their extensive comments and constructive feedback. Addressing the comments:
>
> **Measure of symmetry-invariance of the policy**
>
> To better understand this behavior, we compare the actions produced by policies on trajectories and their color permuted counterparts, e.g. comparing the actions produced by $\pi(\tau)$ to $\pi(\phi(\tau))$. Here $\phi$ is a randomly selected color permutation (the only true symmetry in Hanabi). For level 5 of the hierarchy in pure SP (averaged over 5 seeds):
>
> - Synchronously trained KLR changes actions $25.61 \pm 0.8$% of the time.
> - Sequentially trained KLR changes actions $30.94 \pm 1.9$% of the time.
>
> For reference OP changes actions $16.56\pm 0.17$% of the time. This analysis is quite noisy since ‘changing actions’ could correspond to small changes in the Q-values when there are multiple near equally good actions.
>
> To account for this, we compare the difference in Q-values of the action the agent would normally take on a trajectory to the Q-value of the same color permuted action under the color permuted trajectory, in other words:
>
>
> $ |Q(\arg\max(Q(\tau))|\tau) - Q(\phi(\arg\max(Q(\tau))) |\phi(\tau))|$, where $\phi$ is the color-permutation according to “other-play.”
>
>
> For this analysis we find:
>
> - Synchronously trained agents have a total delta of $24.53 \pm 0.35$ per game.
> - Sequentially trained agents have a total delta of $38.80 \pm 3.41$ per game.
> - For comparison, OP agents have a total delta of $10.72\pm 0.65$ per game.
>
> These analyses indicate that synchronously trained KLRs are quantifiably more color invariant than sequentially trained KLRs. Additionally, as expected, since OP is explicitly trained to be color invariant, it is more color invariant than either KLR.
>
> These findings further support our hypothesis that synchronous training regularizes the KLR, avoiding arbitrary symmetry breaking and improving ZSC.
>
>
> **SyKLRBR vs level 6 of KLR**
>
> We retrained a new hierarchy with 8 levels and we present the results below:
>
>
> | **KLR Level**       | **Cross-Play Score**          | **w/ Rank Bot**          | **w/ Color Bot**        | **w/ Clone Bot**       |
> |-------------|---------------|---------------|----------------|----------------|
> |       6         | $21.80 \pm 0.18$ |   $16.52\pm 0.19$   |  $18.57 \pm 0.26$  |  $15.63 \pm 0.25$   |
> |       7         | $22.45 \pm 0.04$ |   $15.47 \pm 0.44$  |  $19.47 \pm 0.29$  |  $16.18 \pm 0.11$   |
> |       8         | $22.11 \pm 0.11$ |    $15.02 \pm 0.35$  |  $19.23 \pm 0.32$  |  $14.80 \pm 0.29$  |
>
> We notice that although we can increase clone bot score to $16.18$ at level 7, it’s still below that of SyKLRBR. Additionally, the XP scores also appear to level off. As a result, SyKLRBR does provide benefits on top of a vanilla KLR. Lastly, SyKLRBR can be run with a KLR of 7 levels to likely further boost performance.
>
> **Lambda Definition**
>
> Given the Poisson distribution PMF of  $\frac{\lambda^k * e^{-\lambda}}{k!}$ we chose $\lambda = 1$.
>
> **CH results**
>
> In our paper we note that the CH is trained to be a best response to a mix of levels below. This mixture of levels makes providing hints and interpreting hints significantly less stable, which likely leads to poor performance. However, again, if we were to extend this hierarchy to higher levels where each level gains more stability we’d expect the hierarchy to continue to improve.

---

> > ### Author Response · Authors · 2021-08-18
> > **Following up with Reviewer VvuS**
> >
> > We again thank you for taking the time to review. We wanted to follow up with the reviewer to see if they had any more questions/feedback. Additionally, as a summary of our initial response:
> >
> > 1. Synchronous training results in policies that are more color invariant than sequential training.
> > 2. SyKLRBR is distinct from KLR. SyKLRBR with 5 levels performs better in cross-play, ad-hoc teamplay, and with clone-bot when compared with a higher level KLR (levels 6, 7, and 8).
> >
> > Please let us know if you have any more questions.
> >
> > Best,
> >
> > Authors of Paper10963

---

> > > ### Comment · Reviewer_VvuS · 2021-08-23
> > > **Followup to review response**
> > >
> > > Re: the symmetry-invariance, thanks for running this analysis, I think it's compelling and informative.
> > >
> > > Re: SyKLRBR, thanks as well. There is clearly some variance between runs, since table 2 from the paper has higher XP and CloneBot performance for k=5 than the rerun does for k=6. Reporting standard error across multiple runs would make it clearer whether the training run that produced table 3 was just lucky. In absence of this I think the claims about SyKLRBR should be weakened a bit.
> > >
> > > Re: $\lambda$ and the CH model, based on your response to reviewer Y1va, I mostly understand now what you're doing. It would be worth clarifying that your usage is significantly different from the Poisson CH model in [3]: while your level-k is best-responding to a level k-j where j is drawn from Poisson($\lambda$) (truncated somehow, your response didn't specify), in [3] level k best-responds to a level j drawn from Poisson($\lambda$) truncated to $\\{0, \dots, k\\}$.
> > >
> > > Your modification may negatively impact performance, since it means that during training, the possible opponents of level k (i.e. level k-1, level k-2, etc) make more disparate/incompatible assumptions, because they in turn are optimized to best-respond to more differring opponent distributions. In contrast, in the Poisson CH model in [3], the opponents are more stable/consistent, potentially making it more feasible to finetune the strategy. However, the choice of $\lambda$ would then be quite important, and $\lambda=1$ would not result in a very clever agent. Further, as both you and [3] point out, the random level-0 policy may be too weak to make sense to include in the mixture; maybe a shifted Poisson distribution (e.g. where levels k=1 and k=2 are the same as for KLR, and k>2 best-responds to a mix of lower levels not including 0) would be better suited.
> > >
> > > In summary, I think the Poisson CH model would be worth trying as a potentially superior human model, but I'm not sure the existing work in Section 6.6 can be justified. I'd recommend removing it from the paper to improve focus and clarity.

---

> > > > ### Author Response · Authors · 2021-08-25
> > > > **Reply to Reviewer VvuS**
> > > >
> > > > Re variance in XP and Clone Bot: we apologize if there was any confusion around this, but the results in our paper and in our prior response were averaged over 5 seeds and we report standard errors of the mean. As a result, luck shouldn’t be attributed to the results presented in table 3.
> > > >
> > > > Re $\lambda$ and CH: First, for clarity for a given level $k$ in the CH, j is truncated from $\{1, k\}$.
> > > >
> > > > Secondly, Section 2.3 of Cramer et al. 2004 notes that it is common for $\lambda \in (1, 2)$. Under the assumption that level k is a best-response to a level j drawn from $Poisson(\lambda)$ truncated from $(1, ..., k)$, the bulk of the agents will come from lower levels. However, in Hanabi lower levels of the hierarchy do poorly e.g. in KLR level 1 has SP scores of $3.95 \pm 0.28$ and level 2 has a SP score of $14.97 \pm 0.31$. This is likely because they don’t know how to interpret hints well or properly provide hints. We expect that the suggested sampling will lead to worse performance for the given number of levels trained.
> > > >
> > > > To avoid placing too much weight on these lower levels, we adopted the sampling from k-j instead for our implementation of CH from Cramer et al. Thank you for pointing out the differences compared to the original work.
> > > >
> > > > We will try to rerun the experiments with the original sampling before the end of the discussion period and report back the results here. We will also remove the current results of our adaptation of Cramer et al (6.6) from the next version of the submission, as suggested.

---

> > > > > ### Comment · Reviewer_VvuS · 2021-09-08
> > > > > **Followup to response**
> > > > >
> > > > > Regarding variance across 5 seeds: yes, sorry, that was an oversight on my part. Although on the other hand, looking at the cross-play, I think the standard errors seem generally lower than for the other kinds of evaluation (self-play, or vs other bots). There's a bit of subtlety around estimating them: given 5 seeds, there's 20 pairs of seeds on which to do cross-play, but the resulting performances aren't independent samples from the underlying hyperpopulation. This could be addressed using a two-way ANOVA model.
> > > > >
> > > > > Regarding $\lambda\in(1,2)$, I agree that this wouldn't work well with level 0 being random play. One approach to resolving this would be to pick a higher $\lambda$. Another would be to use a different level 0 policy, such as $\ell$-level best response for some $\ell$.
> > > > >
> > > > > Thanks for the clarifications and adjustments.

---

### Official Review · Reviewer_Y1va · 2021-08-01

**Rating:** 6
**Confidence:** 5

**Summary:**

This paper proposes a new solution for the zero-shot coordination problem in Hanabi. Specifically, the authors took the k-level reasoning algorithm and made the training synchronous instead of sequential. Their proposed modification SyKLRBR also trains the agent at level ‘k’ to learn to generate best response to all the previous level players. Authors show that their proposed algorithm has better ad-hoc performance and human coordination than other methods.

**Ethical Concerns:**

None.

**Limitations And Societal Impact:**

Yes.

**Main Review:**


In terms of novelty, this paper does not propose any new algorithm. It takes an existing idea from game theory and tweaks it a bit so that it can be applied to large scale problems like Hanabi. However, the authors have executed this idea very well and I will not penalize the paper for the lack of novelty. There is value in scaling up existing ideas as well.

While I am impressed with the results in this paper, the writing of this paper needs to be improved.

1. I am not able to get a clear idea of what is SyKLRBR doing and how it is different from Synchronous KLR. Figure 1 is very confusing and it is not helping. Is it that only the last level agent learns to play with all the lower-level agents? While I understand Seq KLR and Sync KLR, I need a clear explanation for what Synch CH and SyKLRBR are doing. Pseudocode for all these algorithms in the appendix will help.

2. Introduction and related work have a lot of sentences that are easy to understand only during the second read. This is mainly because the authors keep mentioning levels and they introduce k-level reasoning only later in the paper.

3. Nekoei et al. 2021 also solve the problem of zero-shot coordination for Hanabi and is not mentioned in the related work. Can you provide me with a qualitative comparison of their method and yours? I know quantitative comparison during rebuttal is not plausible due to time constraints, but I highly recommend you to add this comparison for the final version or the next version of this paper.

4. Section 6.6: A best response to a Poisson sample of lower levels. What do you mean by that? With the given explanation, one cannot implement and reproduce your results.

Overall, writing is a major issue with this paper. The writing is vague at some points and hence I cannot reimplement and reproduce results in this paper. I ideally would like to read the next version of this paper before I strongly accept this paper.

References:


[1] Nekoei et al. 2021. Continuous Coordination As a Realistic Scenario for Lifelong Learning.


Minor comments:

1. Line 19 -> did you mean “multi-player” games such as starcraft?
2. Line 141 -> solving solving


**Time Spent Reviewing:**

4

---

> ### Author Response · Authors · 2021-08-11
> **Reply to Reviewer Y1va**
>
> We thank the reviewer for their constructive comments and feedback. We address comments below:
>
> **1. Clarifying CH and SyKLRBR**
>
> We apologize for the clarity of Figure 1 and we will clean it up for the next version of the paper. Clarifying Synch CH, each level $i$ is a best response to a Poisson distribution of levels below. Concretely, this means when generating samples we have 6400 games being played at a time. Each game has a training agent and an agent from a set level below. We use a poisson distribution with a PMF of $\frac{\lambda^k * e^{-\lambda}}{k!}$ and $\lambda = 1$, so for a level $j < i$, $k = i-j-1$. This means that at level 5 ~37% of the actors are from level 4, ~37% are from level 3, ~18% of the actors are from level 2, ~6% are from level 1, and ~1% are from level 0.
>
> For SyKLRBR we synchronously train a KLR and a BR to the KLR. The BR is trained to a Poisson distribution, with $\lambda=1$, of all levels of the KLR hierarchy. Again the Poisson PMF is $\frac{\lambda^k * e^{-\lambda}}{k!}$, where $k = i-j$, for a hierarchy of $i$ levels and a given level $j$. This means that a BR to a 5 level KLR has ~37% of the actors from level 5, ~37% from level 4, ~18% from level 3, ~6% from level 2, ~1% from level 1, and < 1% from level 0.
>
> **2. Comments on Introductions and Related Work**
>
> We appreciate the feedback on the paper and we’ll introduce levels and KLR earlier in the paper.
>
> **3. Qualitative Comparison to Nekoei et al. 2021**
>
> Nekoei et al. use life long learning and multi-task learning to address ZSC. Their method takes a policy (possibly pretrained) and trains it sequentially with some partners. We’d like to note that their best ZSC score is 20.90 (intra-ZSC is equivalent to our definition), which is significantly lower than the SyKLRBR score of 23.29. Additionally, their method indirectly requires access to the symmetries: it samples from a pool of agents, some of which were trained with “other-play”, which in turn requires ground-truth access to the symmetries of the environment. In contrast, our method never utilizes underlying game symmetries and doesn’t require simulator access.
>
> **4. Clarification of Poisson Sampling**
>
> We have clarified the Poisson sampling in 1.
>
> **Minor comments:**
>
> Thank you for pointing these out. For line 19 we did mean to write multi-player games. We will fix these comments in the next iteration.

---

> > ### Author Response · Authors · 2021-08-18
> > **Following up with Reviewer Y1va**
> >
> > We again thank you for taking the time to review. We wanted to follow up to see if the reviewer had any more questions/feedback. Additionally, as a summary of our initial response:
> >
> > 1. We provided clarifications for how to re-implement the Poisson distribution as it appears in the paper.
> > 2. We provided a qualitative comparison to Nekoei et al. 2021, where their best ZSC score of 20.90 is significantly lower than our SyKLRBR score of 23.29. Additionally, their method indirectly requires access to symmetries while our method does not.
> >
> > Please let us know if you have any more questions.
> >
> > Best,
> >
> > Authors of Paper10963

---

### Decision · Program_Chairs · 2021-09-27

**Decision:**

Accept (Poster)

**Comment:**

The authors revisit classic ideas and show that with small modifications they can be made to achieve state-of-the-art coordination with held-out partners in Hanabi. Additionally, the authors provide some insight into why their modifications work, namely in curbing overfitting. While there were some concerns about novelty, the (mostly well-presented) results speak for themselves and make an interesting contribution to the ad-hoc teamplay / zero-shot coordination literature.

That said, I have serious concerns with the current discussion of "human-AI coordination" that I expect to be addressed for the camera-ready. The current draft repeatedly refers to demonstrating "human-AI coordination", however this claim is misleading, since evaluation is done only with a human proxy behavioral cloning (BC) agent, and not actual humans. Although the authors refer to previous circumstantial evidence showing agent scores with BC bots are similar to that with actual humans, this does not imply that this should be true of all agents, and seems an especially dangerous claim for coordination games. Moreover, using human proxies over actual humans removes the possibility of qualitative feedback from human players, as well as subjective preferences over agents, both of which are important components of evaluating human-AI coordination. Indeed, my confidence in the authors claim of BC-human equivalence is further weakened by claims in the draft that their results "confirm that ZSC [aka cross-play (XP)] is a great proxy setting for human-AI coordination and ad-hoc team play." While there seems to be a correlation in all score types, this claim is overblown. For example, in table 3, OP and synchronous KLR achieve similar cross-play scores, but the latter gets nearly double the score of the former with the human proxy - that's a big difference that would be missed if only doing cross-play! If the match between evaluation with human proxies and real humans is as tenuous, the human-AI coordination statements could be worse than misleading and actually wrong. Thus, I expect to see the human-AI coordination claims removed throughout the draft, including section titles, and clarified to specifically refer to coordination with human proxies or BC bots. Moreover, I would like to suggest that the authors in the future do not double down on coordination with human proxy bots alone, since this may well lead to a false sense of progress on their stated goal of coordination with real humans.

One more minor point of feedback - while the set of baselines included is fairly strong and appropriate, there are a couple more that would make the paper's claims stronger. One is simultaneous training but with a fully connected interaction graph. While the authors evaluate a few different interaction graphs, they don't seem to evaluate this very simple one, as far as I can tell. Another are BRs to the various bots (especially to the human proxy), which would help place in context how well the evaluated agents are performing. Finally, as pointed out by Reviewer VvuS, there are issues with the current CH baseline; however, the authors seem aware and are working on fixing them.

Finally, a few more minor points on presentation: Multiple reviewers found figure 1 confusing. This should be easy to improve. Also, I personally found figure 2 to be an odd way to represent probability distributions. The line segments connecting probabilities here are meaningless. Plotting points on a simplex, for example, might be more clear. Finally, [1] is a relevant study on different interaction graphs in multi-agent RL that e.g. should likely be cited for the claims on lines 64 and 351.

[1] Garnelo et al, Pick Your Battles: Interaction Graphs as Population-Level Objectives for Strategic Diversity, AAMAS 2021